# Complement and Coagulation System Crosstalk in Synaptic and Neural Conduction in the Central and Peripheral Nervous Systems

**DOI:** 10.3390/biomedicines9121950

**Published:** 2021-12-20

**Authors:** Shani Berkowitz, Joab Chapman, Amir Dori, Shany Guly Gofrit, Nicola Maggio, Efrat Shavit-Stein

**Affiliations:** 1Department of Neurology, The Chaim Sheba Medical Center, Ramat Gan 5266202, Israel; shanib2@mail.tau.ac.il (S.B.); jchapman@tauex.tau.ac.il (J.C.); amir.dori@sheba.health.gov.il (A.D.); shany.gofrit@sheba.health.gov.il (S.G.G.); nicola.maggio@sheba.health.gov.il (N.M.); 2Department of Neurology and Neurosurgery, Sackler Faculty of Medicine, Tel Aviv University, Tel Aviv 6997801, Israel; 3Department of Physiology and Pharmacology, Sackler Faculty of Medicine, Tel Aviv University, Tel Aviv 6997801, Israel; 4Robert and Martha Harden Chair in Mental and Neurological Diseases, Sackler Faculty of Medicine, Tel Aviv University, Tel Aviv 6997801, Israel; 5Talpiot Medical Leadership Program, The Chaim Sheba Medical Center, Ramat Gan 6997801, Israel; 6Sagol School of Neuroscience, Tel Aviv University, Tel Aviv 6997801, Israel

**Keywords:** synaptic transmission, node of Ranvier, thrombin, neuroinflammation, C1q, C3, stroke

## Abstract

Complement and coagulation are both key systems that defend the body from harm. They share multiple features and are similarly activated. They each play individual roles in the systemic circulation in physiology and pathophysiology, with significant crosstalk between them. Components from both systems are mapped to important structures in the central nervous system (CNS) and peripheral nervous system (PNS). Complement and coagulation participate in critical functions in neuronal development and synaptic plasticity. During pathophysiological states, complement and coagulation factors are upregulated and can modulate synaptic transmission and neuronal conduction. This review summarizes the current evidence regarding the roles of the complement system and the coagulation cascade in the CNS and PNS. Possible crosstalk between the two systems regarding neuroinflammatory-related effects on synaptic transmission and neuronal conduction is explored. Novel treatment based on the modulation of crosstalk between complement and coagulation may perhaps help to alleviate neuroinflammatory effects in diseased states of the CNS and PNS.

## 1. Introduction

The complement and coagulation systems were first described hundreds of years ago [1,2]. At the start of the 20th century, Morawitz first proposed “the classic theory of blood coagulation” [3], while references to the complement system date back to the 19th century [2]. Scientific and medical advances paved the way to establish classical roles for both systems [1,2].

In the past few decades, in addition to their historical functions, local expression of complement and coagulation in the central nervous system (CNS) and the peripheral nervous system (PNS) support additional activities. Each system individually plays a major role in inflammatory processes, with significant crosstalk between them [4,5,6]. An additional player connecting the coagulation and complement systems is the platelet. Platelets, neurons, and glia express common proteins, receptors, and inhibitors attributed to the coagulation system [7,8] and are an important factor for complement pathway activation [9,10]. Intact nerve structure along with a normal environment is critical for physiological synaptic transmission and neuronal conduction. In pathological states occurring in diseases of both the CNS and PNS, there is an increase in proinflammatory mediators that modulate nerve function [11,12]. Aberrant activity of complement and coagulation can lead to neurological deficits [13,14], but how these systems interact together on the synaptic microlevel needs to be elucidated further.

This review summarizes the current evidence regarding the roles of the complement system and the coagulation cascade in the CNS and PNS and presents a plausible hypothesis describing the interplay between them regarding neuroinflammatory-related effects on synaptic transmission and neuronal conduction. 

## 2. The Complement and Coagulation Systems in Physiological States

The complement and coagulation systems are part of the body’s defense mechanism and are involved in the protection from foreign antigens and wound healing, respectively [15,16,17]. The complement system is activated by three main pathways: the classical pathway, the lectin pathway, and the alternative pathway. The three main pathways have critical components that bind with C5b to form the membrane attack complex (MAC) [18]. The activated complement system leads to opsonization, lysis of pathogens, chemotaxis, and inflammation. Anaphylatoxins C3a and C5a are key inflammatory proteins [19]. C3a binds mast cells, causing histamine release and increasing vascular permeability [15]. 

Thrombin, a serine protease, is a main player in the coagulation cascade. Factor (F) Xa activates thrombin by the proteolytic cleavage of the precursor prothrombin [20]. Thrombin mediates the conversion of fibrinogen to fibrin, the main constituent of a blood clot. Additionally, thrombin activates FV, FVIII, FXI, FXIII, and the anticoagulant protein C (PC). Thrombin exerts its physiological function through soluble targets of the coagulation system proteins and its cellular effects through G-protein-coupled receptors. These protease-activated receptors (PARs) belong to a family of seven transmembrane domain receptors, activated through a cleavage process of the extracellular N-terminus [21,22]. PAR1, thrombin’s main receptor, translates the shifts in the proteolytic microenvironment into cellular signaling [23]. Thrombin cleaves this receptor at the extracellular domain to form a new tethered ligand with the sequence thrombin receptor activation peptide (SFLLRN) [24]. PAR1 is alternatively cleaved by activated PC (aPC) bound to its endothelial cell protein C receptor (EPCR), a phenomenon termed biased agonism. This mode of activation is linked to protective and anti-inflammatory responses [25]. 

The complement and coagulation systems share common characteristics. The protein synthesis of both systems takes place mainly in the liver and includes dozens of specific protein components; the coagulation pathways consist of around 20 major proteins while the complement system is made up of more than 40 [26,27,28]. Many of the components are zymogens. The two systems are complex and are activated under specific circumstances [2,29]. Both have additional distinct roles mediated by unique and specific membrane receptors. 

The interaction between the two systems is well established in the periphery and systemic circulation. Recently, accumulated data have indicated that crosstalk between the intrinsic pathway of the coagulation system and the complement system occurs during several conditions, such as changes in pH levels. At the site of inflammation or infection, local pH can fall below six, a change which is sensed by components of the intrinsic pathway of the coagulation system, leading to the indirect activation of the complement system [30]. Each system has components that can activate the other system. Complement can increase tissue factor (TF) activity, thus activating the extrinsic coagulation pathway and forming activated thrombin [31,32,33]. Coagulation can induce the activity of complement factors as well [4,34]. Thrombin, human FIXa, FXa, and FXIa, and plasmin were all found to cleave C3 and C5, and thrombin, in particular, can initiate the activation of C5a in the absence of C3 in- vivo and ex- vivo [34,35] (Figure 1).

Coagulation and complement are both tightly regulated by inhibitors that modulate their respective activity. Natural inhibitors of coagulation factors include antithrombin III, protein S, and PC. Activation of these proteins prevents the activity of specific clotting factors, which provides a regulatory mechanism controlling the coagulation response and limiting clot production [36]. In parallel, complement activity is controlled by soluble or membrane-bound inhibitors such as protein factor H and C4b-binding protein, and cellular receptors such as CD46, CD55, and CD59. CD59 is the main inhibitor of MAC activity. These inhibitors limit spontaneous complement activity and contribute to the termination of response [37]. Crosstalk between coagulation and complement regarding the inhibitory effects is seen as well. The C1 protease inhibitor targets C1 and affects coagulation cascade factors, including FXI, thrombin, plasmin, and tissue plasminogen activator (tPA) [38]. Plasminogen and its activated form plasmin both have complement inhibitory properties in- vitro [39]. Proteolytic cleavage of C5 by thrombin has been reported to create untraditional intermediate products. These, accompanied by the activity of the C5 convertase, assemble a potent MAC [40]. However, these observations are not supported by in- vivo results [41]. The membrane scaffold of the thrombomodulin/thrombin complex activates the aPC/EPCR/PAR1 pathway and inactivates C5a via the thrombin activatable fibrinolysis inhibitor (TAFIa), linking coagulation and complement component interaction [42]. This supports the complexity of the interface between the two systems. 

In the CNS, complement proteins are generated by neurons, microglia, astrocytes, and oligodendrocytes [43]. The thrombin pathway induces cellular processes that can be protective or detrimental, depending on the dose, the receptor mode of activation, and downstream signaling [44,45]. The source of thrombin may be extrinsic, related to inflammatory processes and blood–brain barrier (BBB) breakdown, or intrinsic, mainly secreted by glial cells [46,47]. PAR1 is found throughout the CNS and PNS, mainly in microglia, astrocytes, and oligodendrocytes, and Schwann microvilli at the node of Ranvier (NOR), as well as in neurons and within the blood vascular system [48,49,50]. Thus, many of thrombin’s functions are mediated by glial cells. Classical PAR1 signaling in astrocytes is functionally coupled to Gα_i/o_, Gα_12/13_, and Gα_q/11_ while in neurons signaling is mainly via Gα_q/11_ [51,52]. Astrocytic PAR1 can stimulate ERK phosphorylation, tyrosine kinase activity, and increase intracellular Ca^2+^ [53]. In endothelial cells, thrombin mediates PAR1 to activate ERK via G_i_ [54], and aPC may protect the endothelium from prothrombotic processes [55]_._ In platelets, PAR activation by thrombin initiates multiple signaling cascades by directly coupling mostly to Gα_q_ and Gα_12/13_ [56]_._ Unlike endothelial cells, in human platelets, activated PAR1 undergoes minimal internalization [57]. Recently, C4a was found to be an endogenous agonist ligand that uniquely directly binds to PAR1 and PAR4 in human endothelial cells. C4a activation of PAR1 and PAR4 induces ERK phosphorylation through a Gα_i_-independent signaling pathway. Like other PAR agonists, C4a-mediated stimulation induced a Ca^2+^ increase via the PAR1/Gα_q_/PLCβ signaling axis [58]. In contrast, C4a was not found to act as an agonist in platelets [59]. This may suggest that C4a differentially activates the PAR1 cellular linked pathways rather than the coagulation-based effects in platelet aggregation. This new evidence reinforces the idea that these pathways interact through shared components in the CNS and PNS. 

During development, complement components are localized and expressed by neurons, astrocytes, and glial cells, supported by both mRNA expression and immunohistochemistry [60]. Complement plays a crucial role in synaptic pruning. Less active or “weak” synapses are tagged and removed to allow for stronger and more mature connections [61]. The early complement classical pathway (C1q, C3, and C4) is implicated in mediating synaptic pruning by microglia in the developing retinogeniculate pathway in mice [62,63] and may have a neuroprotective role [64]. Moreover, the complement inhibitory protein SRPX2 was found to be expressed in neurons of the developing mouse brain and participates in the synapse elimination process [65]. The presence of both complement components and inhibitors at basal levels in normal conditions suggests that a fine balance is significant for maintaining a functional role in physiology. 

Thrombin and its receptor PAR1 are key players in synaptic transmission and plasticity [66]. PAR1 in the brain affects synaptic transmission and plasticity by increasing N-methyl-D-aspartate receptor (NMDAR) currents [67,68]. Activation of neurons by thrombin through PAR1 in the hippocampus can lower the epileptic threshold and cause hyperexcitability [68]. Additional experiments support the dose-dependent effects of thrombin on synaptic transmission. High thrombin levels prevent neurons from exhibiting long-term potentiation (LTP) while low levels promote a voltage-gated calcium channel metabotropic glutamate-dependent LTP through the activation of PC [69].

## 3. The Complement and Coagulation Systems in Pathophysiology 

During inflammatory disease states, the complement and coagulation systems can transform from their protective role to a destructive mode through a variety of mechanisms [70,71,72]. Trauma to the CNS and PNS results in the increased permeability of the blood–brain and nerve barriers and a large influx of complement, coagulation components, and immune cells into the neural tissue [19,71,72,73,74]. Animal models of mild traumatic brain injury (mTBI) and Alzheimer’s disease (AD) suggest that there is local production of complement and coagulation components by glial cells including microglia, astrocytes, and neurons [60,75]. Increased complement production, such as C1q, C4b, C3d, C3b, and C5b-C9 terminal complement, is seen in pathophysiological states [60,76]. The lack of C1q in an AD mice model is neuroprotective, supporting complement involvement in AD pathophysiology [77]. Coagulation proteins and inhibitors in plaques were found in multiple sclerosis (MS) patients [78,79] and its animal model experimental autoimmune encephalomyelitis (EAE) [80]. Additionally, the loss of C3 prevented synapse elimination in the hippocampus in mice with EAE [81]. In a mouse schizophrenia model, C4a overexpression reduced cortical synapse density, higher levels of synaptic pruning, altered social behavior, and spatial working memory deficits [82]. These findings strengthen the association between the two systems in CNS and PNS pathophysiology, but the nature of the interrelation remains unknown.

Sepsis-induced disseminated intravascular coagulation (DIC) represents an extreme pathology, in which massive activation of complement and coagulation takes place [83]. Cognitive impairment, a known sequela of sepsis, poses a significant health burden in sepsis survivors [84,85]. Reduced component factors such as C5a, together with elevated levels of inflammatory cytokines, were found to be related to poor cognitive results following sepsis [86]. Future research into pharmacological intervention in the complement cascade may protect against cognitive impairment. 

Abnormal complement activation can be seen in AD patients. Patients deteriorating from minimal cognitive impairment status to AD have elevated levels of astrocyte-derived exosomes containing factors of both classical and alternative complement pathways, including C1q, C4b, C5b, C3b, C5b–C9, and factor D [87]. Expression of C1q in microglia from both AD patients and patients with minimal cognitive impairments is higher compared to healthy controls [88]. One may hypothesize that the early elevated expression of C1q in patients with minimal cognitive impairment results in the later elevation of complement-containing astrocyte-derived exosomes, as described in AD patients. If that is the case, perhaps there is a place for complement manipulation in patients with minimal cognitive impairment, to prevent deterioration to AD. As mentioned above, platelets serve as a complement activation platform. Platelets contain amyloid precursor protein (APP) [8], a known player in AD pathogenesis [89] and a potent coagulation proteases inhibitor [90]. Further characterization of abnormal complement sources in platelets and glial cells will aid in the evaluation of future targeted therapies. 

In mTBI human patients, astrocyte-derived exosome levels of complement components from all three pathways are significantly elevated within the days following mTBI compared to controls [76]. Likewise, coagulation components such as fibrinogen and fibrin depositions are found in human brains after TBI [91], as well as elevated levels of thrombin activity and PAR1 [92]. In stroke, platelets positive for complement proteins were associated with a more severe outcome [93]. Aside from their role in plaque instability, platelet involvement in stroke activates leukocytes, creating an inflammatory environment [94]. Platelets’ ability to activate complement suggests complement activation as another mechanism by which they participate in stroke. 

In the damaged or diseased brain, epitopes exposed by cellular injury, including myelin, are highly vulnerable to complement recognition, opsonization, and MAC deposition [16]. Evidence from DBA/2J mice, a congenital experimental glaucoma animal model, points to C1q having a significant role in open angle glaucoma, a neurodegenerative disorder, and the primary cause of blindness worldwide [95,96]. Neuroinflammation of the optic nerve and retinal ganglion cell layer play a significant role in disease progression [97]. C1q is relocalized at the synapses in the retinas during the early stages of glaucoma and before a significant synaptic loss and retinal ganglion cell death [62]. Following systemic inflammation extrinsically induced by intraperitoneal injections of lipopolysaccharide, hippocampal complement C3 levels in astrocytes and C3a receptor expressions in microglia are upregulated. The application of a C3a receptor antagonist reduces CD68 immunoreactivity and improves cognitive function [98].

The inflammatory role of the thrombin pathway on synaptic function has been implicated in a variety of neurological diseases such as AD, MS, diabetes, cerebral ischemia, and stroke [46]. Following brain injury, thrombin and prothrombin effects differ depending on concentration. Low amounts of thrombin and prothrombin may be neuroprotective, while high amounts can cause neuroinflammation and apoptosis [52]. Post ex- vivo ischemia, thrombin activity increases in hippocampal slices and induces ischemic LTP through the activation of PAR1 and NMDARs. Inhibition of either thrombin or PAR1 restores the physiological LTP [99]. Furthermore, activation of PAR1 alters the excitatory synaptic strength and NMDAR inhibition restores this neuronal function [100]. Complement factors play a role in synaptic function in pathophysiological states as well. CD88, the C5a anaphylatoxin receptor, is expressed locally on presynaptic terminals of mossy fibers in the CA3 region of the adult rat hippocampus, possibly highlighting a role in synaptic plasticity [101]. PAR1 KO mice after occlusion of the middle cerebral artery display lower thrombin and plasmin activity, along with smaller infarcts [102]. Likewise, complement factors increase after ischemic strokes such as C1q and C3. Reduced activation of the classical and lectin pathways possibly preserves neuronal density in ischemic stroke [103]. Future research evaluating complement inhibition as an augmentation to coagulation inhibition in selected patients may improve our abilities to help stroke patients. This evidence supports complement and coagulation cascade neuroinflammatory activity at the synapses (Figure 2). 

Coagulation and complement are involved in central and peripheral pathologies. Mice models of amyotrophic lateral sclerosis (ALS) show a five-fold increase in levels of PAR1 mRNA in the cervical spinal cord compared to wild-type mice [104], as well as an increase in complement activity, which correlates with disease progression [105]. Interestingly, high levels of thrombin are found in the brain of an ALS mouse model. Treatment with PAR1 pathway modulation compounds significantly prolongs survival [106]

Gene expression of complement components is elevated following spinal nerve ligation in rats, and the depletion of C3 in particular attenuated the resulting hyperalgesia seen in this animal model [107]. Following spinal cord injury (SCI), the complement system can be both harmful and neuroprotective. After an injury, there is a rapid increase of thrombin, chemokines, cytokines, and complement components in the spinal cord, and PAR1, in particular, was found to be an important mediator of the neuroinflammation following SCI [108,109,110]. Complement C3 reduces neurite growth along with the restriction of axonal regeneration [111]. Interestingly, complement C1q may have neuroprotective effects. C1q bound to myelin-associated glycoprotein modulates axonal growth and guidance in culture and in- vivo after SCI [112]. 

Similar involvement of both systems is seen after peripheral nerve injury. Rat Schwann cell cultures treated with low levels of thrombin or PAR1 agonist peptide release molecules supporting neuronal survival and neurite elongation. High levels of thrombin or PAR1 agonist peptides induce Schwann cells to release factors that inhibit neurite extension and damage their morphology [113]. Following sciatic nerve crush, fibrin deposition reduces the production of myelin proteins in Schwann cells [114], and thrombin is increased along with FXa activity [115]. Specific inhibition of FXa restores motor function [116]. Moreover, complement immunoreactivity is seen in the myelin sheath of the injured nerve after crush injury [117].

In cases of myasthenia gravis, complement activity contributes to a damaged neuromuscular junction (NMJ). The surface area, the number of acetylcholine receptors, and Na^+^ channels are reduced, leading to abnormal transmission [118]. Hirudin, a specific thrombin inhibitor, and protease nexin 1 (PN1), an endogenous thrombin inhibitor, can block thrombin-induced synaptic loss at the NMJ [119]. After acute peripheral nerve injury, activated microglia and synaptic boutons display positive C1q immunoreactivity occurring near motor neurons which may be involved in synaptic disruption [120]. Modifications of complement and coagulation factors and inhibitors at the NMJ may imply a common role in NMJ pathology. These models further support evidence of complement and coagulation components being activated and impacting synaptic transmission and plasticity. 

## 4. Neuronal Conduction Is Affected by the Complement and Coagulation Systems 

Glial cells control myelin thickness. Oligodendrocytes and Schwann cells myelinate axons in the CNS and PNS, respectively. Normal myelin thickness and gaps at the NOR are critical for normal function [121,122]. The action potential is generated at the axon initial segment and then regenerated at the NOR. Increased amounts of Na^+^ channels in the node, the prevention of diffusion of K^+^ channels into the node, and the narrowing of the gap length increase the membrane resistance and lower capacitance [122]. The integrity of the nodal structure is based on a complex connection between proteins such as neurofascin 186 (NF-186), neurofascin 155 (NF-155), Caspr, Ezrin, and gliomedin [122,123,124]. Gliomedin is a glial protein that interacts with NF-186 and NrCAM, two axonal adhesion molecules, at the NOR. Gliomedin participates in Schwann cell–axon interaction and provides structural support for the node [125]. Assembly of the nodal complex depends on NF-186, acting as a boundary and restricting the migration of paranodal loops into nodal areas, and NF-155 restricts nodal proteins in the axolemma [126]. In the CNS, several extracellular matrix proteins connect axons and glia at the NOR, similar to gliomedin [127]. 

Disruption or pathology of any of these structures results in a modified nodal length and can alter sensory perception, cognitive processing, and motor function. Structural changes of the NOR, including increased length, have been observed in several pathologies including MS and its animal model EAE, aging, cerebral hypoperfusion, diabetes, spinal cord injury, and neonatal hyperoxia [121,128]. Structurally, transgenic mice with deficient astrocyte expression have reduced exocytosis, detached adjacent paranodal loops of myelin from the axon, abnormal myelin thickness in the optic nerve, and larger NOR gaps. Functionally, these mice have approximately a 20% reduction in conduction velocity, delayed spike-time arrival in the cortex, and decreased visual acuity [124]. 

Complement and coagulation may share a mutual role in inflammatory processes that affect nerve conduction (Figure 3). Both the complement system and the coagulation cascade increase inflammatory processes and affect nerve conduction, respectively. Components of each are mapped to key structures that impact nerve conduction. Complement activation followed by MAC formation is an important mechanism for neuronal and glial injury in Guillain–Barré syndrome (GBS) and other demyelinating neuropathies [129,130]. In patients with diabetes, microvascular C5b-9 is increased in both biopsies of denervated muscles and nerve biopsies compared to healthy controls [131]. Patients with untreated chronic inflammatory demyelinating polyneuropathy have increased levels of C5a in the serum and CSF and systemic complement activation correlates with the severity of the disease [132]. This supports the role of complement involvement in peripheral neuropathy. The NOR is a primary site of the immune attack. Immunity towards gliomedin induces progressive neuropathy characterized by conduction deficits and demyelination in spinal nerves [133]. In peripheral neuropathy, IgG deposits are associated with MAC deposition [133]. In a similar model, the deposition of IgG and complement products are concomitant with disruption of Na^+^ channel clusters, abnormal node lengths, and limb weakness progression. Nodal molecules disappear in lesions with complement deposition. During recovery, complement deposition decreases, and Na^+^ channels are redistributed at the affected nodes [134]. The application of a complement inhibitor prevents the disruption of Na^+^ channel clusters and rescues complement deposition in an acute motor axonal neuropathy animal model [135]. Treating animals with thrombin inhibitors prevents the Na^+^ channels from dispersing, restores latency to peak visual evoked potentials, and diminishes the visual deficit [124]. Furthermore, anti-NF-186 and NF-155 antibodies have been found in both CNS and PNS demyelinating disorders [136]. 

Experimental evidence highlights the effects of the coagulation cascade on neuronal conduction as well. Thrombin may negatively affect axonal conduction by the proteolysis of NF-155 [124,137]. Perinodal astrocytes regulate this mechanism by secreting PN1. Deletion of the thrombin binding site on NF-155 results in dysmyelination, paranodal loop dysfunction, nodal gap enlargement, loss of paranodal septate junctions, and misplacement of Caspr1 and nodal Na^+^ channels [137]. The thrombin/NF-155 interaction may not be the only explanation for the coagulation protein’s effects on nerve conduction, as can be seen in several diseases including GBS, diabetic neuropathy, and nerve injuries. Nerve conduction blocks occur upon specific activation of PAR1, which is manifested as compound muscle action potential reductions [49]. The application of a specific PAR1 antagonist completely blocks the negative effects of the PAR1 agonist on nerve conduction [49]. Thrombin levels are increased in the sciatic nerve in experimental autoimmune neuritis and diabetic neuropathy animal models. NOR morphology is damaged and nerve conduction velocity is impaired. Animals treated with a nonspecific thrombin inhibitor, or with a novel PAR1 modulator, have normalized conduction velocity and NOR structure [138,139]. The involvement of complement at the NOR can be seen in diseases such as GBS, which calls for pharmacological intervention in the complement cascade. Indeed, eculizumab, a C5 inhibitor, was evaluated as a possible GBS treatment in a preliminary study [140]. Due to its potential early role in GBS pathogenesis, further study is needed regarding the use of eculizumab as part of the available treatments in GBS, especially early during the disease course, and in combination with selective PAR1 modulation. 

## 5. Summary

Increasing evidence supports key roles of complement and coagulation in the neuroinflammatory damage induced following stroke, neoplasms, epilepsy, traumatic brain and nerve injury, and neurodegenerative diseases [73,141,142]. Coagulation and inflammation are well-known to interact with each other in both physiological and pathophysiological states of the nervous system [14,46,113,143]. As described above, both the complement and coagulation systems participate in neural physiological processes. Complement and coagulation both have significant roles in synaptic transmission and plasticity [65,66] and neuronal conduction [124,129]. The transient inhibition of key pathways in either system hint at novel modulatory techniques [144,145,146]. One of the clinical implications of the complement–coagulation overlap may be that anticoagulants and thrombin receptor antagonists can be beneficial for the neurological manifestations of neuroinflammatory diseases, and not only because of their direct effect on thrombosis. Insights regarding the association between coagulation and complement in the context of neuromodulation hold a promise for future research and treatments. 

## Figures and Tables

**Figure 1 biomedicines-09-01950-f001:**
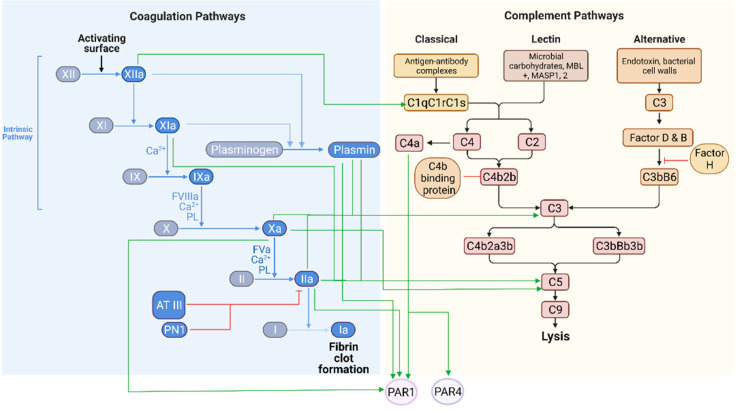
Crosstalk between complement and coagulation: the connections between the coagulation and complement cascades. Complement can increase tissue factor activity, thus activating the extrinsic coagulation pathway and forming activated thrombin. Coagulation can induce the activity of complement factors as well. Thrombin, human FIXa, FXa, and FXIa, and plasmin were all found to cleave C3 and C5, and thrombin can initiate the activation of C5a in the absence of C3 in- vivo and ex- vivo. C4a was found to be an endogenous ligand for PAR1 and PAR4 in human endothelial cells. Antithrombin III (AT III); Protease nexin 1 (PN1); Protease-activated receptor (PAR); Platelet membrane phospholipid (PL); Mannose-binding lectin (MBL); MBL associated serine proteases (MASP). Green arrows indicate activation whereas red lines indicate inhibition. Illustration created with BioRender.com. Accessed date: 16 December 2021.

**Figure 2 biomedicines-09-01950-f002:**
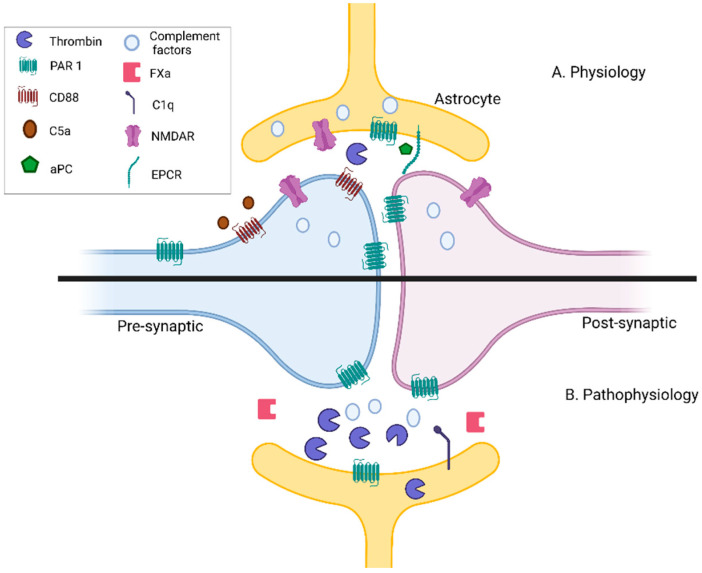
Complement and coagulation cascade neuroinflammatory effects on synapses: (**A**) During development, complement and coagulation components, which are localized to neurons, astrocytes, and glial cells, help mediate synapse elimination in critical pathways in the central nervous system (CNS). C1q increases neuronal survival and arborization as well. CD88, the C5a anaphylatoxin receptor, is expressed locally on presynaptic terminals of mossy fibers in the CA3 region of the adult rat hippocampus. Thrombin and protease-activated receptor 1 (PAR1) are key players in synaptic transmission and plasticity. PAR1 increases N-methyl-D-aspartate receptor (NMDAR) currents, thereby modulating synaptic function. Activated protein C (aPC), endothelial cell protein C receptor (EPCR). (**B**) During pathophysiological states, complement and coagulation components are upregulated and have detrimental effects on synaptic transmission and plasticity. Reactive astrocytes express C1q, as well as other complement proteins, which may affect synaptic loss in the adult CNS. Following neuroinflammation, the thrombin pathway impacts synaptic function in neurological diseases such as AD, MS, diabetes, cerebral ischemia, and stroke. Illustration created with BioRender.com. Accessed date: 16 December 2021.

**Figure 3 biomedicines-09-01950-f003:**
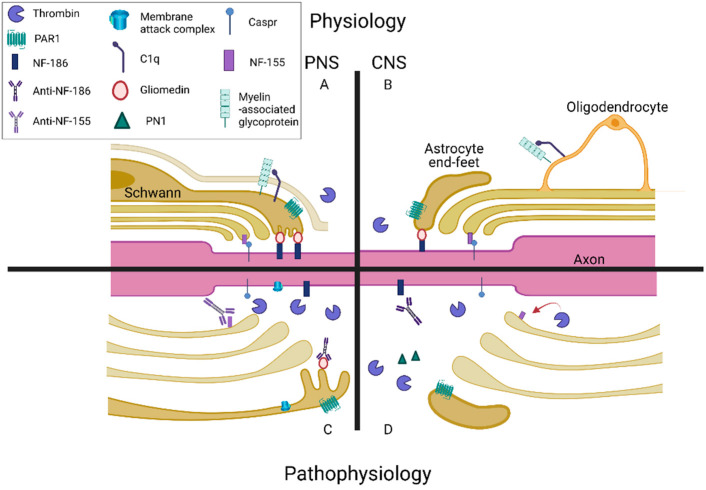
Complement and coagulation cascade neuroinflammatory effects on the node of Ranvier (NOR): (**A**) Peripheral nervous system (PNS) NOR: Schwann cells express protease-activating receptor 1 (PAR1) at the microvilli. Neurofascin-186 (NF-186) interacts with gliomedin in the matrix and in the microvilli to promote axon–Schwann cell microvilli attachment. Neurofascin-155 (NF-155), a paranodal protein, acts as a cell adhesion molecule between axons and myelin. In physiological conditions, complement and coagulation factors are downregulated in the PNS. C1q is bound to myelin-associated glycoprotein (MAG), a transmembrane glycoprotein localized in Schwann cells and oligodendrocytes. (**B**) Central nervous system (CNS) NOR: Normal myelin thickness and gaps at the NOR are mediated by astrocyte exocytosis. PAR1 is localized on the cell body and astrocytic endfeet. Thrombin is generated by neuronal and glial cells. (**C**) PNS NOR pathology: Complement activation and membrane attack complex (MAC) formation are upregulated in pathophysiological conditions. MAC mediates the cell-killing effect of the complement cascade. The NOR is a primary site of immune attack. Anti-NF186 and NF-155 antibodies have been found in PNS demyelinating disorders. In the PNS, thrombin levels increased in diseased states. NOR morphology was damaged and nerve conduction velocity was impaired. (**D**) CNS NOR pathology: A plausible role of the complement system as a part of the coagulation–inflammation interface is suggested. Thrombin proteolysis of NF-155 has negative effects on axonal conduction. Inhibitors of thrombin activity such as protease nexin 1 (PN1) are locally expressed in the brain. Perinodal astrocytes regulate this mechanism by secreting PN1. Illustration created with BioRender.com. Accessed date: 16 December 2021.

## Data Availability

Not applicable.

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
