# Peer review of "Complement and Coagulation System Crosstalk in Synaptic and Neural Conduction in the Central and Peripheral Nervous Systems"

_biomedicines, 2021, doi:10.3390/biomedicines9121950_

Round 1

Reviewer 1 Report

It is a very interesting and comprehensive review related to the important role of complement and blood coagulation systems in neurologic diseases.

General comment

It would be important, in my opinion, to include some information (several sentences) related to the role of platelets in the regulation of blood coagulation, complement activation, and functions of neurons in formal and diseased CNS. Also, more information should be included to review related to the role of complement (C1, C3, C4) in synapse pruning.

Specific comments:

  1. All figures should have a higher resolution
  2. Legend for Fig.1: More information should be included in the legend regarding crosstalk between blood coagulation and immune systems.
  3. Page 3, line 113. The sentence "Many of thrombin's functions...." is not clear. Please re-write to make it clear and more specific.
  4. Page 4, line 132, The sentence "Activation of thrombin through PAR..." is not clear. Thrombin activates PAR in neuronal cells etc. Please make it clear.
  5. Could you discuss in more detail signaling in neurons downstream of PAR receptors? Want would be similarities and differences in PAR signaling in platelets vs. neurons vs. glia? Why signaling through PAR is different for endothelial cells vs. platelets in the case of non-classic PAR agonist C4a? Would be great if the authors include some hypotheses or thoughts about this.

Author Response

Dear Reviewer 1,
Thank you for reviewing our manuscript titled “Complement and coagulation system crosstalk
in synaptic and neural conduction in the central and peripheral nervous systems”. We greatly
appreciate your positive feedback. Below is our point-to-point response in blue to each of your
comments. Changes in the text are written in blue. We hope that the issues you have clarified
for us have been addressed and we have edited the manuscript accordingly. We hope that the
current form of the article meets your high standards and is suitable for publication in
“Biomedicines.
Thank you for this opportunity,
Efrat Stein-Shavit, on behalf of the authors
Review 1
It is a very interesting and comprehensive review related to the important role of complement and
blood coagulation systems in neurologic diseases.
General comment
It would be important, in my opinion, to include some information (several sentences) related to the
role of platelets in the regulation of blood coagulation, complement activation, and functions of
neurons in formal and diseased CNS. Also, more information should be included to review related to
the role of complement (C1, C3, C4) in synapse pruning.
We thank the reviewer for this important comment. Additional information regarding the role of the
platelet as a connection between complement and coagulation, as well as its resemblance to
neurons, was added to the introduction section (page 1 lines 43-47) and to the pathophysiology
section (page 5, lines 208-212). We have also added some more information relating to the role of
complement (C1, C3, C4) in synapse pruning in physiology (page 4 lines 156-160) and in
pathophysiology (page 5 lines 186-189).

Specific comments:
1. All figures should have a higher resolution
The figures were edited as per your suggestions, the text and icons were enlarged as well.
2. Legend for Fig.1: More information should be included in the legend regarding crosstalk
between blood coagulation and immune systems.
More information regarding crosstalk between blood coagulation and complement factors was
added to the legend of Figure 1 to obtain optimal clarity (page 3, lines 101-106).
3. Page 3, line 113. The sentence "Many of thrombin's functions...." is not clear. Please re-write
to make it clear and more specific.
The sentence was moved for clarification and context (page 4, line 137).
4. Page 4, line 132, The sentence "Activation of thrombin through PAR..." is not clear. Thrombin
activates PAR in neuronal cells etc. Please make it clear.
The sentence was edited to achieve clarification and emphasize the activation of neurons
(page 4, line 167).
ï‚· Could you discuss in more detail signaling in neurons downstream of PAR receptors? Want
would be similarities and differences in PAR signaling in platelets vs. neurons vs. glia? Why
signaling through PAR is different for endothelial cells vs. platelets in the case of non-classic
PAR agonist C4a? Would be great if the authors include some hypotheses or thoughts about
this.

A few additional sentences regarding the downstream signaling of PAR receptors were added
(page 4, line 137-145) as well as additional sentences regarding signaling via C4a (page 4,
line 145-153).

Reviewer 2 Report

The authors prepared a scholarly, timely and convincing review highlighting cross-talk between intermediates of the complement and coagulation cascades, which support their physiologic and pathogenic involvement in the peripheral and central nervous systems.  

I am wondering if a sentence has been omitted between the end of the line numbered 282 on page 7 of 13 and beginning of line 303 on page 8 of 13 of the pdf?

Given the balanced involvement of some intermediates in protection against insult/injury/inflammation and their roles in pathogenesis, which often appear to depend on timing and amount of expression, I am wondering if the authors have speculated about practical translational therapeutic implications of their review and research?  If so, I am sure the readers would appreciate a brief summary of their thinking. 

Author Response

Dear Reviewer 2,
Thank you for reviewing our manuscript titled “Complement and coagulation system crosstalk
in synaptic and neural conduction in the central and peripheral nervous systems”. We greatly
appreciate your positive feedback. Below is our point-to-point response in blue to each of your
comments. Changes in the text are written in blue. We hope that the issues you have clarified
for us have been addressed and we have edited the manuscript accordingly. We hope that the
current form of the article meets your high standards and is suitable for publication in
“Biomedicines.
Thank you for this opportunity,
Efrat Stein-Shavit, on behalf of the authors
Review 2
The authors prepared a scholarly, timely and convincing review highlighting cross-talk between
intermediates of the complement and coagulation cascades, which support their physiologic and
pathogenic involvement in the peripheral and central nervous systems.
I am wondering if a sentence has been omitted between the end of the line numbered 282 on page 7
of 13 and beginning of line 303 on page 8 of 13 of the pdf?
Line 343 on page 8 (formerly 282) was edited for clarification.
Regarding lines 374-376 on page 9 (formerly line 303 on page 8): thank you for your good eye. The
sentence has been updated for completion.
Given the balanced involvement of some intermediates in protection against insult/injury/inflammation
and their roles in pathogenesis, which often appear to depend on timing and amount of expression, I
am wondering if the authors have speculated about practical translational therapeutic implications of
their review and research? If so, I am sure the readers would appreciate a brief summary of their
thinking.
This is a very important point, and we thank the reviewer for raising it. A paragraph related to the
possible effect of complement in sepsis-induced cognitive impairment was added to page 5 lines 192-
198, followed by a paragraph related to Alzheimer’s disease (page 5, lines 199-208). Additionally, a
few lines describing stroke were added (page 5, 217-221). A phrase regarding the possible use of the
C5 inhibitor eculizumab in GBS treatment was added to the nerve conduction section (page 11, lines
389-394)

Reviewer 3 Report

In this manuscript, the authors have the ambitious project to review complement and coagulation system functions, interplay in the neurological system. Since the subject is wide, most of the topics are described in a very concise way but the main focus is to connect complement and coagulation system to cnetral and peripheral neurological disorders.

Line: 71: SFLLRN: fullname please

Line 112-115: need references, or your readers may naturally think that these from your own comments.

Line 124: you need to clarify  whether complement components are generated or localized to neurones............................... If neurones can generate and the complements are localized to neurons, how and why the neurons damage do not occur daily?

Line 153-154: syntax error (no verb).

Line 162: Althoug I am not the ophthalmologist, I do knwo that there are at least two types of glaucoma, open-angle and closed angle. The underlying for these two are totally different. Based onyour statement, do you suggest that C1q have a significant role in both types of glaucoma?  I don't think so. You need to read https://www.sciencedirect.com/science/article/pii/S1350946220300884

Please summarize this paper in your MS.

line 165: following systemic inflammation? extrinsic or intrnsic induced?

Line 174: confused with line 135. Please be consistent.

Line 177: you try to introduce CD88, but what is the relationship between CD88 and the previous sentence?

FIgure 2: the explanation of aPC and EPCR did not appear in your context. What is the role of these two factors in the crosstalk between complement and coagulation system?

Line 240: of Ranvier "(NOR)"

Line 243-250: need to rewritte . What is the role and function of gliomedin?  you should integrate line 272 with 243-250

line 272: The NOR ...............(need references) 

line 303: 155????

Summary is relatively weak, did not clearly and completely reflect what you were trying to show in this work.

I like the line 324to 327. You should put these lines in the beginning of the summary.  Then tell your readers what's wrong with complements and coagulation system in the diseases. How you or people try to solve ..................In this way, your work will be more attractive.

Author Response

Dear Reviewer 3,
Thank you for reviewing our manuscript titled “Complement and coagulation system crosstalk
in synaptic and neural conduction in the central and peripheral nervous systems”. We greatly
appreciate your positive feedback. Below is our point-to-point response in blue to each of your
comments. Changes in the text are written in blue. We hope that the issues you have clarified
for us have been addressed and we have edited the manuscript accordingly. We hope that the
current form of the article meets your high standards and is suitable for publication in
“Biomedicines.
Thank you for this opportunity,
Efrat Stein-Shavit, on behalf of the authors
Review 3
In this manuscript, the authors have the ambitious project to review complement and coagulation
system functions, interplay in the neurological system. Since the subject is wide, most of the topics
are described in a very concise way but the main focus is to connect complement and coagulation
system to cnetral and peripheral neurological disorders.
Line: 71: SFLLRN: fullname please
The full name was added as per your suggestion (page 2, line 76).
Line 112-115: need references, or your readers may naturally think that these from your own
comments.
References were added to clarify that these were not our comments (page 4, lines 131-133).
Line 124: you need to clarify whether complement components are generated or localized to
neurones............................... If neurones can generate and the complements are localized to
neurons, how and why the neurons damage do not occur daily?
Additional clarification was added to elucidate the local expression of the complement components to
neural cells (page 4, lines 154-156) and a suggested hypothesis regarding complement in physiology
was added (page 4, lines 162-164).
Line 153-154: syntax error (no verb).
The error was edited (page 4, line 182).
Line 162: Althoug I am not the ophthalmologist, I do knwo that there are at least two types of
glaucoma, open-angle and closed angle. The underlying for these two are totally different. Based
onyour statement, do you suggest that C1q have a significant role in both types of glaucoma? I don't
think so. You need to read https://www.sciencedirect.com/science/article/pii/S1350946220300884
Please summarize this paper in your MS.
We thank the reviewer for emphasizing the difference and for the informative review. We speculate
that C1q is involved in the neuroinflammatory aspect of open-angle glaucoma (page 5, lines 224-226)
and have added additional information from the suggested manuscript above (page 5, lines 226-228).

line 165: following systemic inflammation? extrinsic or intrnsic induced?
The clarification that this was following extrinsic induced systemic inflammation was added
(page 5, line 230).
Line 174: confused with line 135. Please be consistent.
These two sentences describe two different types of long-term potentiation (LTP). The first line
describes the LTP response to different doses of thrombin in normal hippocampal slices while the
second line refers to LTP in hippocampal slices during pathophysiological ischemic conditions.
Line 177: you try to introduce CD88, but what is the relationship between CD88 and the previous
sentence?
An additional sentence was added to emphasize the relationship between the sentences (page 6,
lines 242-243).
FIgure 2: the explanation of aPC and EPCR did not appear in your context. What is the role of these
two factors in the crosstalk between complement and coagulation system?
Thank you for this significant comment. We introduced the aPC/EPCR in the text (page 2, lines 77-79)
and we added important details regarding the aPC/EPCR pathway and its interaction with the
complement system (page 3 lines 126-128).
Line 240: of Ranvier "(NOR)"
Thank you for your note, this was corrected
Line 243-250: need to rewritte. What is the role and function of gliomedin? you should integrate line
272 with 243-250
The role of gliomedin was added (page 7, lines 314-320), and the paragraph was rephrased.
line 272: The NOR ...............(need references)
A reference was added (page 8, lines 341-343).
line 303: 155????
Thank you for catching this. A sentence preceding 155 was missing. It was added (page 9, lines 374-
376).
Summary is relatively weak, did not clearly and completely reflect what you were trying to show in this
work.
I like the line 324to 327. You should put these lines in the beginning of the summary. Then tell your
readers what's wrong with complements and coagulation system in the diseases. How you or people
try to solve ..................In this way, your work will be more attractive.
Thank you for your helpful comment. The summary was edited per your suggestion. The lines you
suggested were moved to the beginning of the summary (page 9, lines 397-399). We also added our
insights regarding the clinical application (page 9, lines 405-408). We hope that it is improved.

Round 2

Reviewer 3 Report

accepted